# Fundamental Study on Hydrogen Low-NOx Combustion Using Exhaust Gas Self-Recirculation

**Kenta Kikuchi \*** , **Tsukasa Hori and Fumiteru Akamatsu**

Department of Mechanical Engineering, Osaka University, Osaka 5650871, Japan; thori@mech.eng.osaka-u.ac.jp (T.H.); akamatsu@mech.eng.osaka-u.ac.jp (F.A.)
* Correspondence: k.kikuchi@tokyo-gas.co.jp; Tel.: +81-3-3803-9804

**Abstract:** Hydrogen is expected to be a next-generation energy source that does not emit carbon dioxide, but when used as a fuel, the issue is the increase in the amount of NOx that is caused by the increase in flame temperature. In this study, we experimentally investigated NOx emissions rate when hydrogen was burned in a hydrocarbon gas burner, which is used in a wide temperature range. As a result of the experiments, the amount of NOx when burning hydrogen in a nozzle mixed burner was twice as high as when burning city gas. However, by increasing the flow velocity of the combustion air, the amount of NOx could be reduced. In addition, by reducing the number of combustion air nozzles rather than decreasing the diameter of the air nozzles, a larger recirculation flow could be formed into the furnace, and the amount of NOx could be reduced by up to 51%. Furthermore, the amount of exhaust gas recirculation was estimated from the reduction rate of NOx, and the validity was confirmed by the relationship between adiabatic flame temperature and NOx calculated from the equilibrium calculation by chemical kinetics simulator software.

**Keywords:** NOx; hydrogen; exhaust gas self-recirculation; burner





## 1. Introduction

In order to reduce carbon dioxide emissions from combustion, the use of green fuels, such as hydrogen, ammonia, and eFuel generated from renewable energy sources, is under consideration. Among them, hydrogen is generated in the coke oven gas produced in steel mills and as a byproduct gas in the production of caustic soda (NaOH) and chlorine gas (Cl$_2$), so there are growing expectations for recovering and using them as energy [1,2]. It is also expected to be used as a so-called energy carrier to transport renewable energy, which is difficult to transport and store, in the form of carbon-free fuels such as hydrogen and ammonia. Hydrogen has a higher burning velocity and lower minimum ignition energy than ammonia, and therefore is a fuel with less risk of misfiring or unburned compared to ammonia [3–6]. On the other hand, hydrogen has a higher flame temperature than ammonia and hydrocarbon fuels, so reducing thermal-NOx is an important challenge [7,8]. There are some reports on the development of industrial burners designed specifically for hydrogen combustion to achieve low-NOx combustion in high temperature conditions, such as aluminum melting furnace [9–11]. However, when hydrogen obtained as a byproduct gas is used as fuel, the supply of hydrogen is not stable, so it is necessary to be able to easily switch to co-firing with hydrocarbon fuels or hydrocarbon fuels only. If a hydrocarbon fueled burner can be modified to enable low-NOx hydrogen combustion with a simple modification, rather than a burner structure specifically designed for hydrogen fuel, it will not only reduce the cost of modifying existing facilities, but also make it possible to easily switch between hydrocarbon and hydrogen fuels. There are many reports for the formation of NOx in hydrogen flame [12–15]. In diffusion combustion under lean conditions at low pressure, such as in industrial furnaces, the dominant route of NO formation is the route produced by the extended Zeldovich mechanism, commonly referred to as thermal-NO [16]. Because of the high activation energy, the reaction of this formation route is

strongly temperature depended: higher flame temperature, higher residence time, and higher oxygen concentration lead to higher NO emission [17] (p. 200).

One method of low-NOx combustion in hydrocarbon fueled burners is the use of exhaust gas recirculation. By circulating the flue gas in the furnace into the combustion air and reducing the oxygen concentration, the flame temperature is lowered and low-NOx combustion is achieved [18]. There are two types of exhaust gas recirculation: the one is the method that uses power such as a blower to circulate the exhaust gas, and the other is exhaust gas self-recirculation, which uses the momentum of the fuel gas and combustion air to circulate the exhaust gas. The latter is often used in industrial furnaces. Exhaust gas recirculation has a trade-off with combustion stability, and if the recirculation rate is set to an excessive amount, such as more than 20%, it can affect the combustibility and cause pulsating combustion or misfire due to blowout [17] (p. 202).

Although there have been reports on the effects of hydrogen combustion in a premixed burner [19–22] and on the effects of exhaust gas recirculation in hydrogen engines [23–25], there have been no studies on the NO formation characteristics of non-premixed hydrogen combustion in an industrial furnace and the effects of exhaust gas recirculation have not yet been studied. Since hydrogen has a very high laminar burning velocity, abnormal combustion such as knocking in gas engines and backfiring in premixed burners such as boilers is a problem [24]. On the other hand, burners in industrial furnaces, where most of the combustion is diffusion at low pressure, do not have such problems. Rather, the large burning velocity and wide flammability range may not cause abnormal combustion even if the EGR rate is set higher than that of hydrocarbon fuels. While exhaust gas self-recirculation is a very useful method for reducing thermal NOx in industrial burners, there is no systematic report on the effect of the nozzle design on the combustion characteristics of exhaust gas self-recirculation. In this study, we experimentally evaluated the rate of increase of NOx when hydrogen was burned in an experimental furnace using the practical burner with simple structure for hydrocarbon fuels, and the effects of changing the diameter and number of combustion air nozzles in order to reduce NOx with simple modifications.

## 2. Experiments and Methods

The configurations of the experimental equipment used in this study are shown in Figures 1 and 2. The inside dimensions of the experimental furnace are W1200 mm × H1200 mm × L3500 mm. The inside of the furnace is surrounded by 260 mm thick ceramic fiber insulation made of alumina and silica (ISOWOOL BSSR1300, Isolite Industries, Osaka, Japan). Three ports for R-thermocouples are installed in the upper part of the furnace, and the ambient temperature at a point 100 mm from the ceiling of the furnace is constantly measured every five seconds. The axial positions of the thermocouples are Z = 250 mm, 1450 mm, 2600 mm, with Z = 0 mm at the furnace wall on the burner side. The thermocouple has a strand diameter of 0.5 mm and is covered with a 15 mm diameter ceramic protection tube. There are 12 tubes for loading at a height of 100 mm above the furnace floor, and each tube is equipped with equal cooling air to control the temperature at the point Z = 2600 mm to the required temperature. There is a sight glass on the back wall of the furnace, directly in front of the burner, so that the flame can be observed. There is an exhaust gas sampling hole at the side wall of the furnace downstream, which is connected to an exhaust gas analyzer (PG-340, HORIBA, Kyoto, Japan). The measurement principles were the magnetic dumbbell method for $O_2$, non-dispersive infrared absorption method for CO, and the chemiluminescence method for NOx, respectively.

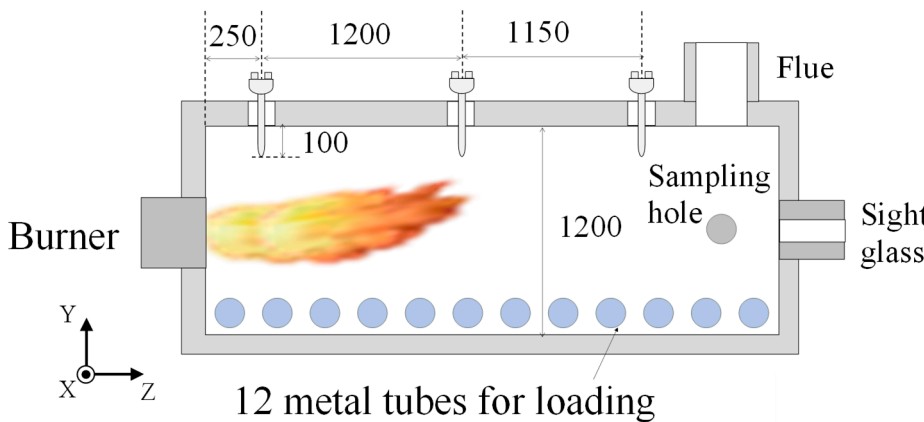

**Figure 1.** Test furnace size.

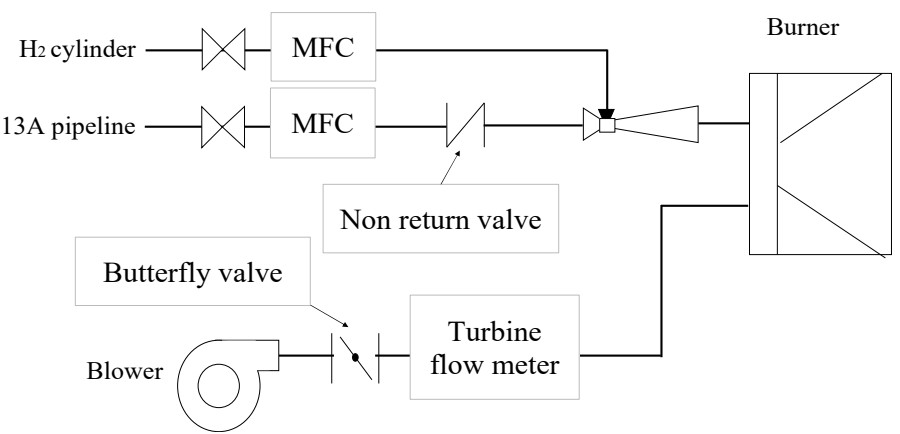

**Figure 2.** Pipe flow of test furnace burner.

We used a diffusion combustion burner of the recirculation flow retention type in this experiments. As shown in Figure 3, this is a non-premixed burner with eight air nozzles on a plate at the end of the fuel supply pipe, which forms a stable flame by inducing part of the fuel gas into the self-recirculating flow formed by the combustion air jetted at high speed. It is one of the most commonly used burners in industrial furnace applications and is also widely used in actual industrial furnaces [26]. In this study, we adjust combustion air velocity by replacing or plugging the air nozzles.

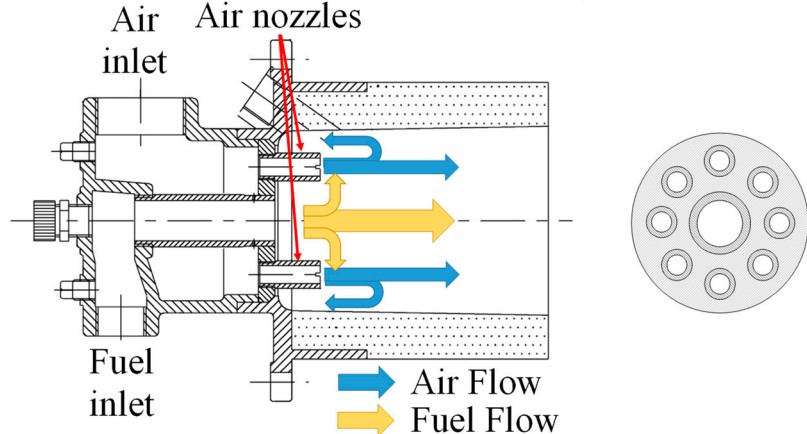

**Figure 3.** The burner structure.

In this experiment, we used a mixture of city gas 13A and $H_2$ as fuel. The typical composition of city gas 13A is $CH_4$ = 89.6%, $C_2H_6$ = 5.62%, $C_3H_8$ = 3.40%, and $C_4H_{10}$ = 1.35% [27]. The lower heating value of city gas 13A is calculated to be 40.63 MJ/Nm$^3$, and that of $H_2$ is 10.83 MJ/Nm$^3$. City gas is supplied from a pipeline, $H_2$ from a cylinder. After being depressurized by a regulator, the mixture is adjusted to the desired ratio by mass flow controllers and supplied at room temperature. Combustion air is supplied from a blower, and the flow rate is manually adjusted by butterfly valve to required flow rate value on the turbine flow meter.

In the experiment, we measured CO under the low temperature condition of 723 K and NOx under the high temperature condition of 1273 K at the point of $Z$ = 2600 mm. Under both conditions, the pressure in the furnace $P$ was adjusted to +15 Pa by the adjustment damper in the exhaust flue and kept constant. The experimental parameters were the ratio of $H_2$ in the lower heating value of the fuel mixture "$E_{H2}$" and excess air ratio "$\lambda$". $E_{H2}$ is given by Equation (1), where the mole fraction of hydrogen in the fuel mixture is $X_{H2}$ and the lower heating values of city gas 13A and hydrogen are $LHV_{13A}$ and $LHV_{H2}$, respectively.

$$E_{H2}\% = \frac{X_{H2} \times LHV_{H2} \times 100}{X_{H2} \times LHV_{H2} + (1 - X_{H2}) \times LHV_{13A}} \tag{1}$$

The air nozzles conditions for the experiments are shown in Table 1. $V_a$ and $V_f$ in the table show the air flow velocity and fuel gas flow velocity at an excess air ratio of 1.2 during hydrogen combustion, respectively. $\varphi11.9 \times 8$ is a city gas model condition. In $\varphi10.2 \times 8$ and $\varphi8.4 \times 8$, the air nozzles were replaced with smaller diameter ones to increase the combustion air velocity by 1.4 and 2 times, respectively. In $\varphi11.9 \times 4$, the air flow velocity is doubled by reducing the number of air nozzles by half instead of the air nozzle diameter.

**Table 1.** Air nozzle conditions.

| No. | Case | Air Nozzles Diameter $D_a$ [mm] | Number of Air Nozzles $N$ | Gas Velocity Ratio $V_a/V_f$ |
|---|---|---|---|---|
| 1 | $\varphi11.9 \times 8$ | 11.9 | 8 | 2.36 |
| 2 | $\varphi10.2 \times 8$ | 10.2 | 8 | 3.21 |
| 3 | $\varphi8.4 \times 8$ | 8.4 | 8 | 4.74 |
| 4 | $\varphi11.9 \times 4$ | 11.9 | 4 | 4.72 |

## 3. Results and Discussion

### 3.1. Combustion Stability

Since hydrogen has a very fast burning velocity and a very low minimum ignition energy, in non-premixed combustion, it rarely causes abnormal combustion, such as pulsating combustion, loss of fire, or generation of unburned content [3,5]. Therefore, we evaluated the combustion stability of each condition based on the amount of CO emitted at 723 K in city gas combustion ($E_{H2}$ = 0%).

Figure 4 shows the upper limit of the excess air ratio "$\lambda$" at which the burner can burn at less than 100 ppm CO at $E_{H2}$ = 0% and 723 K under each condition. In $\varphi11.9 \times 8$ and $\varphi10.2 \times 8$, combustion was able to continue without CO emission up to the condition of $\lambda$ = 1.5 for any combustion rate. However, in $\varphi8.4 \times 8$, CO was emitted over 100 ppm at $\lambda$ > 1.15 and 174 kW of input. The amount of CO increased in $\varphi11.9 \times 4$, and the CO was less than 100 ppm only when $\lambda$ was smaller than 1.05 with a small combustion rate. In addition, at 174 kW, misfires occurred and the combustion was very unstable.

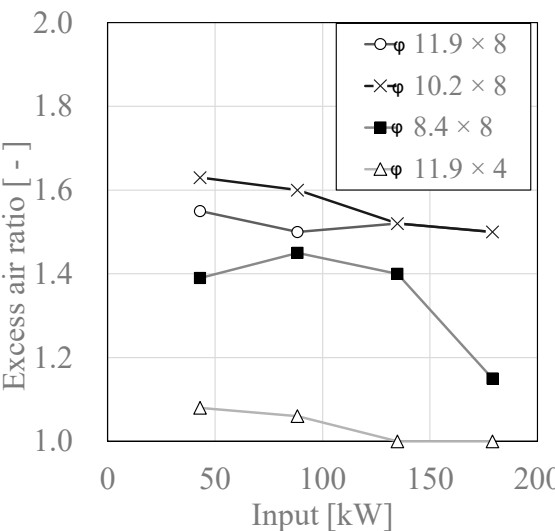

**Figure 4.** Combustion stability range (CO < 100 ppm) for different combustion rates under each air nozzle case, at 723 K, $P$ = +15 Pa and $E_{H2}$ = 0%.

On the other hand, in hydrogen combustion ($E_{H2}$ = 100%), there was no pulsating combustion or misfire, and the combustion was stable and continuous in all cases.

Figure 5 shows the amount of CO at 723 K and $\lambda$ = 1.2 under each case. This figure also shows that a large amount of CO was emitted in $\varphi 11.9 \times 4$. It can also be seen that there was almost no difference in the CO emission characteristics between $\varphi 11.9 \times 8$ and $\varphi 10.2 \times 8$.

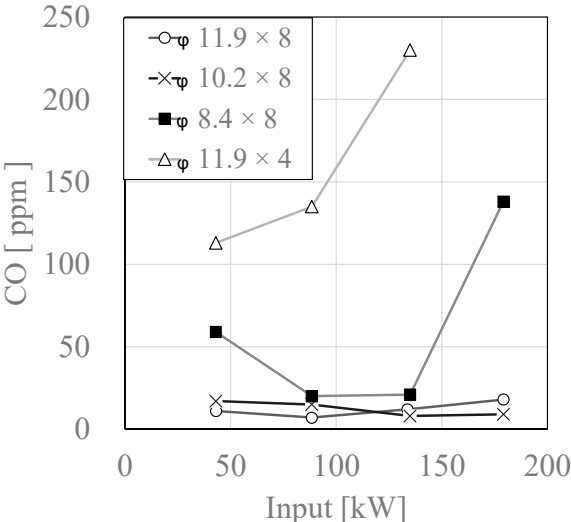

**Figure 5.** CO emissions characteristics for different combustion rates under each air nozzle case. At 723 K, $P$ = +15 Pa, $\lambda$ = 1.2 and $E_{H2}$ = 0%.

Figure 6 shows photographs of the flame at $E_{H2}$ = 0% and 1273 K. In general, in combustion including hydrocarbons, such as city gas, it is known that the blue–green color appears in rich-fuel combustion due to the strong band spectrum around 0.51 µm emitted by $C_2$ radicals, and the blue color appears in lean-fuel combustion due to the strong band spectrum around 0.39 µm and 0.43 µm emitted by CH radicals [28]. In $\varphi 11.9 \times 8$, the entire inside of the burner tile had a light blue flame, which was characteristic of lean combustion, while in $\varphi 8.4 \times 8$ and $\varphi 11.9 \times 4$, the flame was divided into two parts: one part had a clear dark blue flame that could be visually confirmed and the other part had no visible flame. In $\varphi 8.4 \times 8$ and $\varphi 11.9 \times 4$, the flames were unstable, constantly changing the location where the visual flame could be seen.

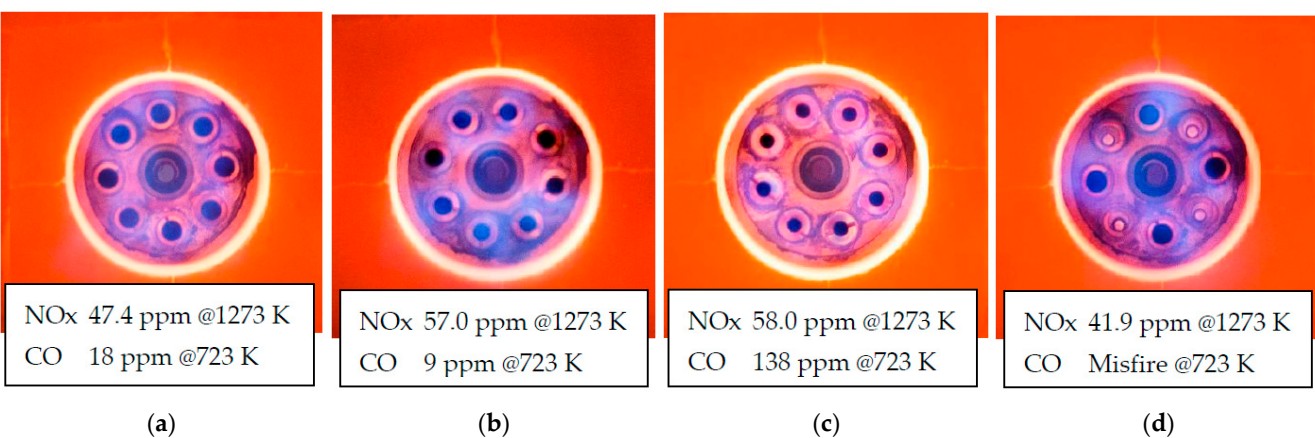

|  |  |  |  |
|---|---|---|---|
| NOx 47.4 ppm @1273 K<br>CO 18 ppm @723 K | NOx 57.0 ppm @1273 K<br>CO 9 ppm @723 K | NOx 58.0 ppm @1273 K<br>CO 138 ppm @723 K | NOx 41.9 ppm @1273 K<br>CO Misfire @723 K |
| **(a)** | **(b)** | **(c)** | **(d)** |

**Figure 6.** Flame photographs at $E_{H2}$ = 0% and 1273 K of (**a**) $\varphi$11.9 × 8; (**b**) $\varphi$10.2 × 8; (**c**) $\varphi$8.4 × 8; (**d**) $\varphi$11.9 × 4. Input = 174 kW, $\lambda$ = 1.2, $P$ = +15 Pa.

Figure 7 shows photographs of the flame at $E_{H2}$ = 100% and 1273 K. In hydrogen combustion, the $C_2$ and CH radicals mentioned above do not exist, and therefore the flame could not be visually confirmed.

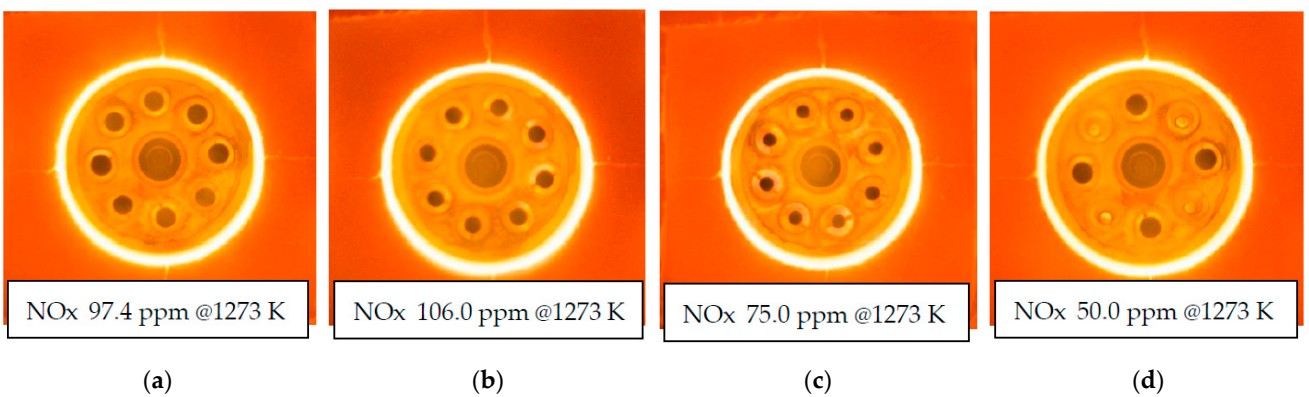

|  |  |  |  |
|---|---|---|---|
| NOx 97.4 ppm @1273 K | NOx 106.0 ppm @1273 K | NOx 75.0 ppm @1273 K | NOx 50.0 ppm @1273 K |
| **(a)** | **(b)** | **(c)** | **(d)** |

**Figure 7.** Flame photographs at $E_{H2}$ = 100% and 1273 K of (**a**) $\varphi$11.9 × 8; (**b**) $\Phi\varphi$10.2 × 8; (**c**) $\varphi$8.4 × 8; (**d**) $\varphi$11.9 × 4. Input = 174 kW, $\lambda$ = 1.2, $P$ = +15 Pa.

### 3.2. Temperature Distribution

Figures 8 and 9 show the temperature distribution in the furnace when the temperature at the point Z = 2600 mm is 1273 K at $E_{H2}$ = 0% and 100%, respectively. From the figures, it can be seen that the temperature tends to increase from the front to the back of the furnace in both $E_{H2}$ conditions. Comparing $E_{H2}$ = 0% and 100%, there was no significant difference in temperature distribution for the same air nozzles condition. Comparing each air nozzles condition, $\varphi$11.9 × 8 condition had the lowest temperature upstream of the furnace in both $E_{H2}$ conditions.

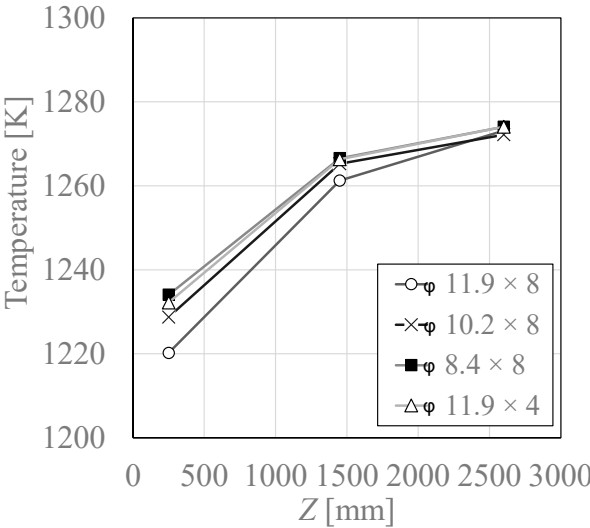

**Figure 8.** Temperature distribution when the temperature at the point $Z$ = 2600 mm is 1273 K at $E_{H2}$ = 0%. Input = 174 kW, $\lambda$ = 1.2, $P$ = +15 Pa.

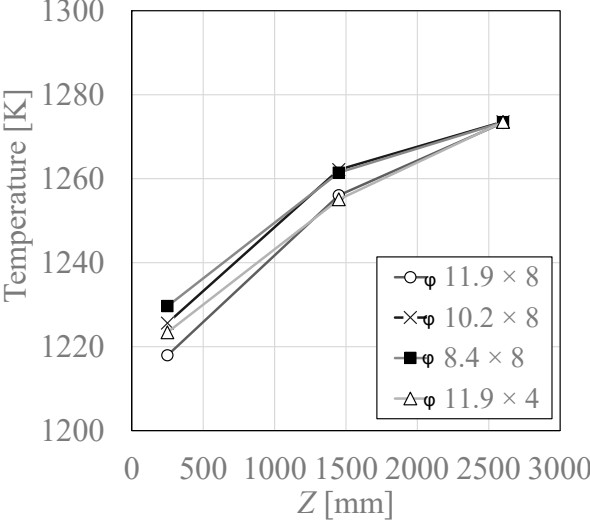

**Figure 9.** Temperature distribution when the temperature at the point $Z$ = 2600 mm is 1273 K and $E_{H2}$ = 100%. Input = 174 kW, $\lambda$ = 1.2, $P$ = +15 Pa.

*3.3. NOx Emissions Characteristics*

Figure 10 shows the relationship between NOx and $E_{H2}$ obtained in a preliminary experiment in another small test furnace at 1273 K. Although the burner structure was different, the characteristics of a rapid increase in NOx was obtained above $E_{H2}$ = 80%, and therefore, we measured NOx by varying $E_{H2}$ in 5% intervals above $E_{H2}$ = 80% in this study.

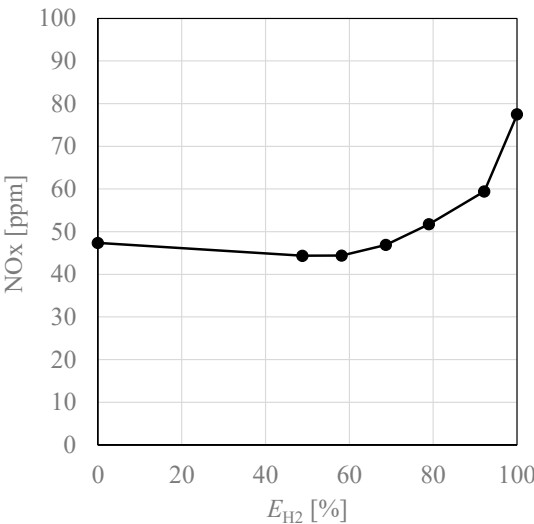

**Figure 10.** NOx in a small test furnace at 1273 K and $P$ = +15 Pa. Input = 11 kW, $\lambda$ = 1.2.

Figure 11 shows the amount of NOx for different $E_{H2}$ under each case. As shown in Figure 11 in φ11.9 × 8, NOx gradually increased from 47.4 ppm in city gas combustion ($E_{H2}$ = 0%) to 97.4 ppm in hydrogen combustion ($E_{H2}$ = 100%). The reason for this is that the adiabatic flame temperature of hydrogen (2382 K) is about 150 K higher than that of city gas (2233 K) [17] (p. 28). In φ10.2 × 8, NOx was higher than in φ11.9 × 8 for all $E_{H2}$ ranges. In φ8.4 × 8, NOx was higher than in φ11.9 × 8 when $E_{H2}$ was small, and reversed when $E_{H2}$ was large, resulting in lower NOx. In φ11.9 × 4, NOx was lower than in φ11.9 × 8 for all $E_{H2}$ ranges. In φ11.9 × 4, we succeeded in reducing NOx in hydrogen combustion to the same level as that in city gas combustion in φ11.9 × 8.

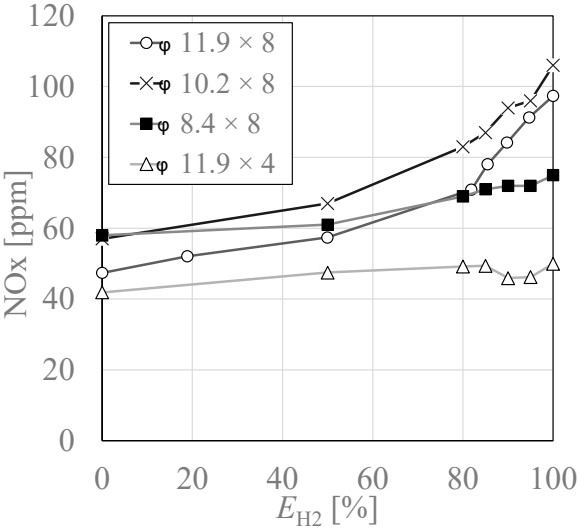

**Figure 11.** NOx for different $E_{H2}$ under each air nozzle case, at 1273 K. Input = 174 kW, $\lambda$ = 1.2, $P$ = +15 Pa.

Figure 12 shows the comparative NOx emission rates for Cases B–D, with φ11.9 × 8 being 1.0. Comparing city gas combustion with hydrogen combustion, it can be seen that NOx was reduced by 20% in φ8.4 × 8 and by 50% in φ11.9 × 4. However, in Φφ10.2 × 8 and φ8.4 × 8, NOx increased by about 20% more than in φ11.9 × 8 during city gas combustion.

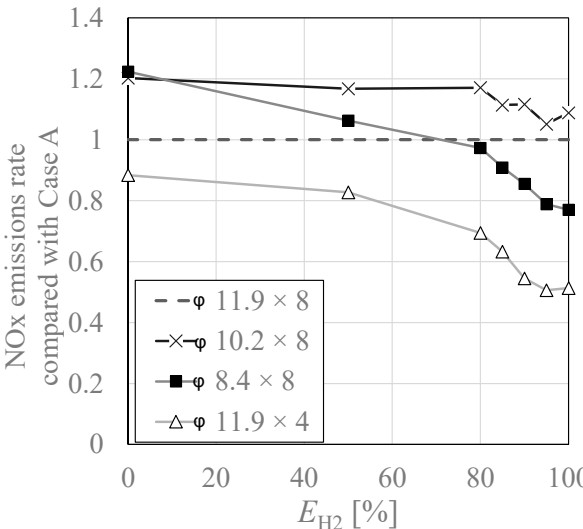

**Figure 12.** NOx emissions rate compared with $\varphi 11.9 \times 8$ for different $E_{H2}$ under each air nozzles case at 1273 K. Input = 174 kW, $\lambda$ = 1.2, $P$ = +15 Pa.

### 3.4. Relationship between Excess Air Ratio "λ" and NOx

Figure 13 shows the relationship between $\lambda$ and NOx in city gas combustion ($E_{H2}$ = 0%). In city gas combustion, NOx increased or remained almost flat as $\lambda$ increased. This tendency is consistent with the generally known NOx characteristics of hydrocarbon fuels [17] (p. 209). Figure 14 shows the relationship between $\lambda$ and NOx in hydrogen combustion ($E_{H2}$ = 100%). In all cases, there was no significant change in the NOx when $\lambda$ was in the range of 1.05 to 1.2. On the other hand, when $\lambda$ was set to 1.4 or higher, the NOx in $\varphi 11.9 \times 8$ and $\varphi 10.2 \times 8$ tended to decrease. This can be said to be a specific characteristic of hydrogen combustion, which was not seen in city gas combustion. Generally, in premixed combustion of hydrocarbon fuels, NOx peaks around $\lambda$ of 1.0, and NOx decreases as $\lambda$ increases [17,29] (pp. 207–209). Since hydrogen has very fast burning velocity, its NOx emissions characteristics in diffusion combustion may also be close to those of premixed combustion.

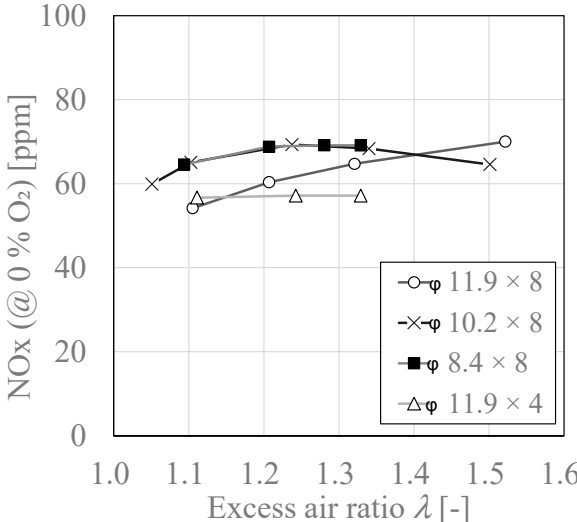

**Figure 13.** NOx for different $\lambda$ under each air nozzle case, at 1273 K and $E_{H2}$ = 0%. Input = 174 kW, $P$ = +15 Pa.

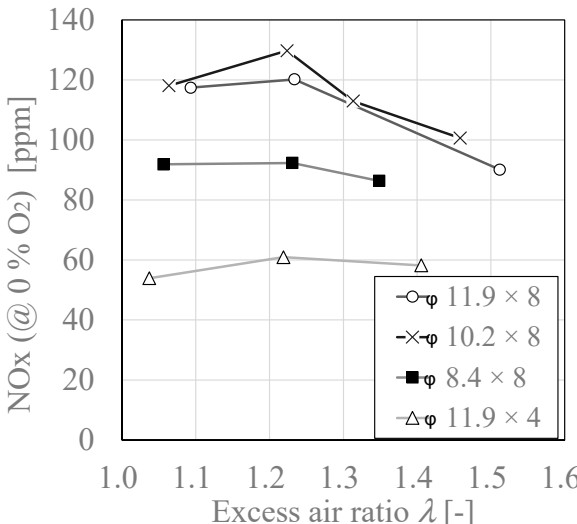

**Figure 14.** NOx for different $\lambda$ under each air nozzle case, at 1273 K and $E_{H2} = 100\%$. Input = 174 kW, $P = +15$ Pa.

### 3.5. Relationship between Adiabatic Flame Temperature "$T_f$" and NOx

Flame temperature is one of the most important properties to consider in thermal NOx emissions characteristics. Adiabatic flame temperature is often used to evaluate the properties of fuels. Figure 15 shows the relationship between NOx and adiabatic flame temperature "$T_f$", which is strongly related to the formation of Thermal-NO. $T_f$ at ambient pressure and temperature was simulated using CHEMIKIN II with changing $E_{H2}$ and $\lambda$. In exhaust gas self-recirculation, it is difficult to measure recirculation rate, so that these adiabatic flame temperatures were simulated without considering the exhaust gas recirculation. As shown in Figure 15, NOx rapidly increased as $T_f$ increased above 2100 K in $\varphi 11.9 \times 8$ and $\varphi 10.2 \times 8$. On the other hand, in $\varphi 8.4 \times 8$ and $\varphi 11.9 \times 4$, NOx increased linearly with increasing $T_f$ and there is no inflection point. Therefore, in $\varphi 8.4 \times 8$ and $\varphi 11.9 \times 4$, the actual flame temperature was considered to be even lower due to exhaust gas recirculation. In addition, the slopes in the graphs of $\varphi 8.4 \times 8$ and $\varphi 11.9 \times 4$ were almost equal. In general, reports on the low-NOx effect of exhaust gas recirculation for hydrocarbon fuels show that NOx was reduced by about 50% and 20% when the exhaust gas recirculation rate (EGR rate) was set to 10% and 5%, respectively [17] (pp. 202–203). As shown in Figure 12, in $\varphi 8.4 \times 8$ and $\varphi 11.9 \times 4$, NOx in $E_{H2} = 100\%$ was reduced by about 20% and 50%, respectively. Therefore, we assumed EGR rates of 5% and 10% for $\varphi 8.4 \times 8$ and $\varphi 11.9 \times 4$, respectively, and re-plotted the adiabatic flame temperature accounting for exhaust gas recirculation, which is shown in Figure 16. It was found that the curves were almost all the same, although the NOx for $\varphi 10.2 \times 8$ and $\varphi 8.4 \times 8$ were slightly higher in $T_f < 2100$ K. This curve shows that the $T_f$ in $\varphi 11.9 \times 4$ was below 2100 K due to exhaust gas recirculation even during hydrogen combustion, which suppressed the increase in NOx. The reason for the higher NOx in $\varphi 10.2 \times 8$ and $\varphi 8.4 \times 8$ when $T_f$ was small ($E_{H2}$ was small) was probably due to the lower local combustion air ratio caused by the larger recirculation flow formed near the air nozzles due to the increase of combustion air velocity, which increased the amount of fuel gas drawn near the air nozzles. The decrease in the local combustion air ratio is expected to increase the flame temperature and lead to an increase in NOx generation. The decrease in the local combustion air ratio is also consistent with the darker blue color in the flame photograph in Figure 6. As $E_{H2}$ increased, the local air ratio bias decreased as the fuel gas velocity increases, since less fuel gas was drawn into the air nozzles.

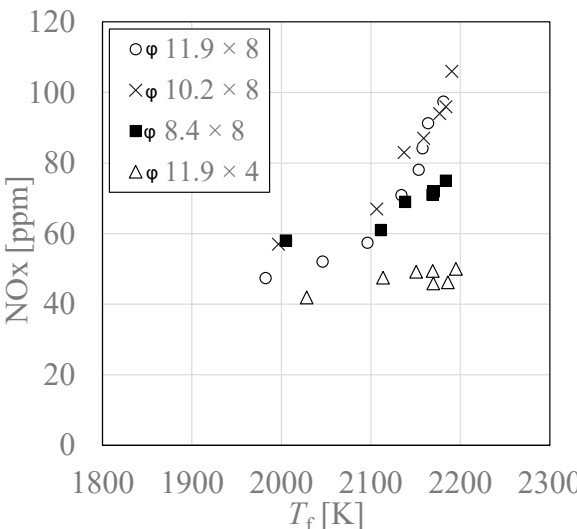

**Figure 15.** NOx for different $T_f$ without accounting for EGR at 1273 K. Input = 174 kW. $\lambda$ = 1.2, $P$ = +15 Pa. $E_{H2}$ = 0, 50, 80, 85, 90, 95, 100%.

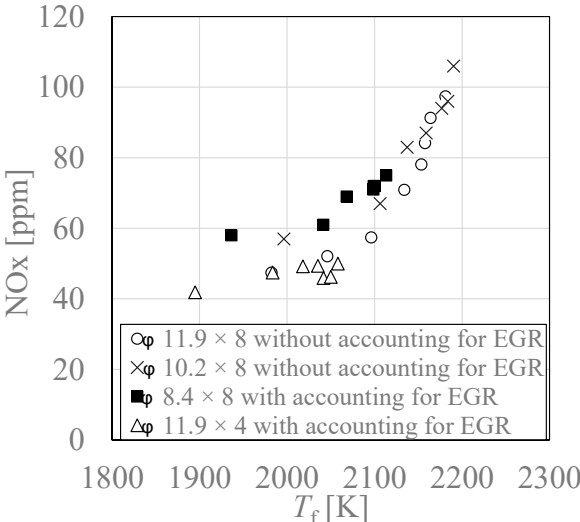

**Figure 16.** NOx for different $T_f$ accounting for EGR at 1273 K. Input = 174 kW. $\lambda$ = 1.2. $P$ = +15 Pa. $E_{H2}$ = 0, 50, 80, 85, 90, 95, 100%.

Comparing $\varphi 8.4 \times 8$ and $\varphi 11.9 \times 4$, the gas velocity ratio $V_a / V_f$ is the same, however, their combustion characteristics were very different. This is due to the different distance between the air nozzles in $\varphi 8.4 \times 8$ and $\varphi 11.9 \times 4$. Figure 17 shows nozzle layout of $\varphi 8.4 \times 8$ and $\varphi 11.9 \times 4$. Defining the distances between the air nozzles in $\varphi 8.4 \times 8$ and $\varphi 11.9 \times 4$ as $L_c$ and $L_d$, respectively, we obtain $L_d / L_c = 1.85$. In general, when multiple jets are placed in a forest, the space closed between the jets becomes depressurized and a recirculation flow is formed there [30]. Jianchun et al. developed an approximate equation that expresses the recirculation rate in terms of various jet parameters, including the distance between jets, where the recirculation rate and the distance between jets are proportional [31]. Therefore, in $\varphi 11.9 \times 4$, the EGR rate was higher, and the oxygen concentration in the combustion air was lower, resulting in unstable combustion to the level that CO was emitted at low temperatures, and it is thought that NOx was reduced by lowering the flame temperature even in hydrogen combustion.

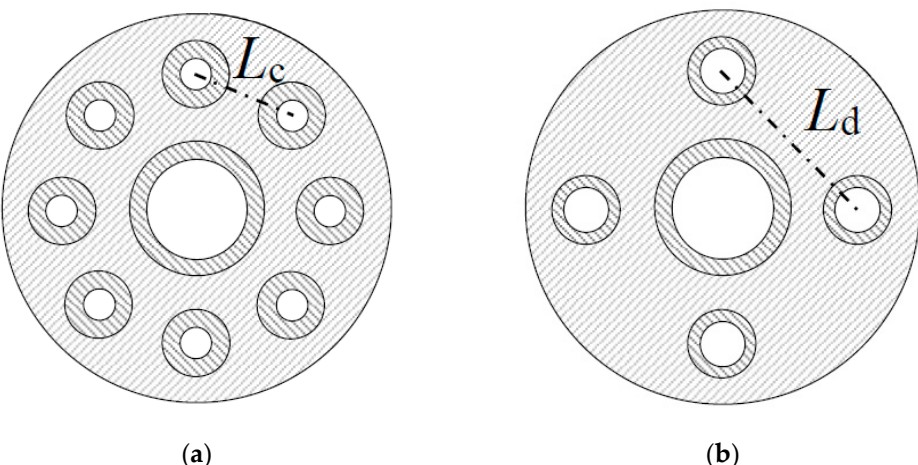

(**a**)                                                                                     (**b**)

**Figure 17.** Nozzle layout of (**a**) φ8.4 × 8; (**b**) φ11.9 × 4.

## 4. Conclusions

We experimentally evaluated the combustion characteristics of exhaust gas self-recirculation combustion and the effect of combustion air nozzle parameters using recirculation flow retention type burner, and examined the possibility of realizing low-NOx hydrogen combustion. The following conclusions can be obtained.

(1)   When hydrogen was burned using a burner with city gas specifications, NOx increased twice as much due to the increase in flame temperature.

(2)   When the air nozzles diameter was reduced and the air flow velocity was increased by a factor of 1.4, the amount of fuel gas drawn into the air nozzle increased and the local air ratio decreased, resulting in an increase in NOx.

(3)   When the combustion air nozzle diameter was reduced and the combustion air flow velocity was doubled, NOx increased under low hydrogen mixing ratio conditions as same as mentioned above, but NOx reduced under high hydrogen mixing ratio conditions due to exhaust gas recirculation.

(4)   When the number of air nozzles was reduced to half instead of the air nozzles diameter and the combustion air velocity was doubled, NOx was reduced at all co-firing rates, and a 50% reduction was successfully achieved in hydrogen combustion. And the NOx in this case was almost equal to the NOx in the case of city gas combustion in the city gas model.

(5)   By reducing the number of air nozzles, the distance between air nozzles became larger, resulting in a larger EGR ratio. As a result, misfire occurred in hydrocarbon combustion, but in hydrogen combustion, stable combustion could be continued without abnormal combustion.

(6)   When the number of air nozzles was reduced to half, the exhaust gas recirculation rate was about 10%, which was assumed from the adiabatic flame temperature simulated by the equilibrium calculation of chemical kinetics simulator software.

(7)   It was confirmed that low-NOx combustion by exhaust gas recirculation is possible in hydrogen combustion.

**Author Contributions:** Conceptualization, K.K.; software, K.K.; validation, K.K.; formal analysis, K.K.; investigation, K.K.; resources, K.K. and F.A.; writing—original draft preparation, K.K.; writing—review and editing, T.H. and F.A.; supervision, F.A.; project administration, F.A. All authors have read and agreed to the published version of the manuscript.

**Funding:** This research received no external funding.

**Acknowledgments:** We would like to express my appreciation to YOKOI KIKAI KOSAKUSYO Co., Ltd. and TAKAMITSU Co., Ltd. for their assistance in the construction of the experimental apparatus.

**Conflicts of Interest:** The authors declare no conflict of interest.

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
