# Peer review of "Fundamental Study on Hydrogen Low-NOx Combustion Using Exhaust Gas Self-Recirculation"

_processes, doi:10.3390/pr10010130_

Round 1

Reviewer 1 Report

The topic seems appropriate for the journal processes and of great interest to the readership. The approach seems valid and the measurement procedure seems to be sensible. The analysis could go a little more into detail and especially the figures need to be extended. The manuscript is well written, but need some improvements before publication.

Major corrections:

  1. Please insert all boundaries in every single figure description: E_H2, lambda, P. One of these may be missing in the description, if it is the x-axis. Add E_H2 even if city gas or hydrogen is mentioned in the caption to improve the clarity
  2. Figure 7 to 14: I would strongly suggest to picture NOx, CO emissions and all three temperatures for all pictures. Especially the temperatures measured by the introduced thermocouples are not mentioned anywhere. Once the figures are extended, please add the resulting information in the text.

Minor revisions:

  1. Line 19: Dispense with abbreviations and brand names in the abstract
  2. Line 29: What is “better” combustion stability? Please clarify. Hydrogen tends to form combustion instabilities and thus to burn significantly faster than the laminar flame speed. This seems to contradict your statement
  3. Introduce the burner application earlier. What is it used for in detail. It should be clearly delaminated from domestic heating burners.
  4. Figure 1: The western reading direct is from left to right. Normally, also pictures follow that direction. Consider mirroring of the burner.
  5. Line 109: Add E_H2 as equation. Maybe it would also be sensible to replace it by the ratio directly
  6. Line 110: It might be sensible to always call it “air nozzle”, to avoid confusion with fuel nozzles as used in other applications
  7. Table 1: Try to choose more descriptive names than Case A / B / C / D. For example, reference (normal air nozzle), medium air nozzle, small air nozzle, Half No. of normal air nozzles.
  8. Line 122/123: I strongly suggest to discuss the statement of the combustion stability for the transfer from city gas to hydrogen operation. I understand that the CO emissions will no longer be applicable. But do you expect the same stability behavior for hydrogen operation? I would not, due to the strongly different burn characteristics. As you mentioned in the introduction, others do fully replace the burner layout to enable stable hydrogen operation.
  9. Figure 4: Fuel needs to be added in the figure caption
  10. Figure 4 / 5: Very poor picture quality
  11. Line 144 - 149: The picture quality is quite poor and I am not sure, whether I can share these observations.
  12. Figure 6: There should be enough room to have all pictures in one row. Why do you not also zoom closer to the burner tiles?
  13. Line 154: Can you explain that stronger radiation? What is the cause for that?
  14. Figure 7: See comments to figure 6
  15. Line 168: I cannot see the flame temperatures in any diagram. I think you should support this statement with data
  16. Line 185: I do not understand the following statement: “NOx is converted to O2=11%”. How can O2 be constant if the excess air ratio changes? Please clarify and add an explanation
  17. Line 193: This does not match my experience. I expect higher NOx emissions at slightly lean conditions (lambda > 1), where more oxygen is present for the NOx formation. Additional literature would be useful, to discuss the NOx emission dependency on the excess air ratio.
  18. Line 215: Why 5 and 10 %? This sounds a little random. Is it just guessed? Could you maybe change the order of figure 14 and 15 and calculate the EGR ratio based on Jianchun et al.?
  19. Line 215: Case A, B and C should be a constant function, right? So I think you should correct Case B as well. E.g. half as much as you correct Case C.
  20. Figure 13/14: The combustion process includes internal EGR in both cases. I with extend the legend to “with EGR correlation” or “with/without accounting for EGR”
  21. Figure 13/14: What is the fuel? E_H2 = ???
  22. Line 256: I would think that the decreasing local air ratio causes increased temperature and this temperature is the origin of the NOx increase. Please provide the data to support/neglect that assumption and explain it more in detail
  23. Line 258 and 266: Some readers start with the Conclusions section. Please introduce abbreviation and tools again or subscribe them with common descriptions.
  24. References: There has been quite some publishing activities in the latest past and a high activity of hydrogen combustion research. Please ensure to include the latest literature, e.g. from Processes and maybe also from Energies and include these findings in your discussion.

Author Response

Thank you for your review of our paper.

We have answered each of your points in the attached file.

We hope the revised version is now suitable for publication and look forward to hearing from you in due course.

Reviewer 2 Report

The topic of the manuscript is of interest to the community because it describes how NOx can be reduced through the nozzle design of the burner for hydrogen and hydrocarbon combustion. The findings are interesting but could benefit from a somewhat more precise description. The general understanding could also be improved by adding the flow pattern including the fundamental flow recirculation to Figure 1 or Figure 3.

Comments:

  • Line 27: I suggest to use the term “energy carrier” for hydrogen.
  • Line 31: It would be interesting to know more about the design of these burners. How do they prevent the formation of NOx?
  • Line 33: It is not fully clear to me why the supply of H2 is intermittent. Can’t it be stored and supplied continuously?
  • Line 49: There is a word missing: “.. the one is the method [that] uses…”
  • Line 81: There is a word missing: ”… at the side wall of [the] furnace…”
  • Figure 1: It would be beneficial to indicate the coordinate system (X,Y,Z) and the direction of flow in the chamber.
  • Figure 2: The quality of the figure is quite low in my version of the manuscript. There is a typo in the figure caption (“sylinder”).
  • Figure 3: The right part of the figure has quite low quality.
  • Figure 4: The figure has quite low quality.
  • Line 135: A word is missing: “… shows that [a] large amount of CO was emitted…”
  • Figure 4: The symbols of Case A and Case C are very similar. Please consider using different symbols.
  • Line 140: Consider rephrasing to e.g. “Figure 6 [shows] photographs of [the] flame …”
  • Figure 6a has quite low quality.
  • Line 185: It is not clear what the conversion of NOx to O2=11% means. Please explain.
  • Figure 11: Please explain the development of NOx with changing excess air ratio. Why does it increase linearly for case A, while it remains more or less flat for all other cases? Does this have to do with the gas velocity?
  • Figure 12: Please explain the development of NOx with changing air excess ratio. Why does the NOx decrease for H2 combustion with rising lambda values for cases A-C and why is it flat for case D? You mention that in premixed combustion of hydrocarbon fuels NOx peaks at lambda=1.0. Can you explain why and why hydrogen combustion is similar to this case?
  • Figure 14: You say in the text that all curves are more or less the same but that cases B and C are slightly higher. I appears they are up to ~50% higher than the other cases, which is significant. Can you explain this further?
  • Figure 15: The figure caption is not centered.
  • Conclusions: (2) Is this not only the case for cases A and B (looking at Figure 11)?
  • Conclusions: (4) Please explain the difference between reducing the nozzle diameter and the number of nozzles. In one case, NOx is reduced and in the other it is increased. This is probably due to a higher recirculation rate in case of case D? Please elaborate.

Author Response

(The authors gave the same response as above.)

Round 2

Reviewer 1 Report

The authors have improved the manuscript and addressed my comments.

Thank you

Reviewer 2 Report

I believe the work on the manuscript justifies publication in its present form.